# Association between the Preoperative *C*-Reactive Protein-to-Albumin Ratio and the Risk for Postoperative Pancreatic Fistula following Distal Pancreatectomy for Pancreatic Cancer

**DOI:** 10.3390/nu14245277

**Published:** 2022-12-10

**Authors:** Naotake Funamizu, Kyosei Sogabe, Mikiya Shine, Masahiko Honjo, Akimasa Sakamoto, Yusuke Nishi, Takashi Matsui, Mio Uraoka, Tomoyuki Nagaoka, Miku Iwata, Chihiro Ito, Kei Tamura, Katsunori Sakamoto, Kohei Ogawa, Yasutsugu Takada

**Affiliations:** Department of HBP Surgery, Ehime University, 454 Shitsukawa, Toon 791-0295, Japan

**Keywords:** *c*-reactive protein-to-albumin ratio, distal pancreatectomy, postoperative complications, postoperative pancreatic fistula

## Abstract

Postoperative pancreatic fistula (POPF) are major postoperative complications (POCs) following distal pancreatectomy (DP). Notably, POPF may worsen the prognosis of patients with pancreatic cancer. Previously reported risks for POCs include body mass index, pancreatic texture, and albumin levels. Moreover, the *C*-reactive protein-to-albumin ratio (CAR) is a valuable parameter for prognostication. On the other hand, POCs sometimes lead to a worse prognosis in several cancer types. Thus, we assumed that CAR could be a risk factor for POPFs. This study investigated whether CAR can predict POPF risk in patients with pancreatic cancer following DP. This retrospective study included 72 patients who underwent DP for pancreatic cancer at Ehime University between January 2009 and August 2022. All patients underwent preoperative CAR screening. Risk factors for POPF were analyzed. POPF were observed in 17 of 72 (23.6%) patients. POPF were significantly associated with a higher CAR (*p* = 0.001). The receiver operating characteristic curve analysis determined the cutoff value for CAR to be 0.05 (sensitivity: 76.5%, specificity: 88.9%, likelihood ratio: 6.88), indicating an increased POPF risk. Univariate and multivariate analysis revealed that CAR ≥ 0.05 was a statistically independent factor for POPF (*p* < 0.001, *p* = 0.013). Therefore, CAR has the potential to predict POPF following DP.

## 1. Introduction

Distal pancreatectomy (DP) is the standard surgical procedure for tumors located in the pancreatic body or tail, such as pancreatic cancer, neuroendocrine neoplasm, and mucinous cystic neoplasm [1]. A postoperative pancreatic fistula (POPF) is one of the most serious postoperative complications (POCs) of DP. Despite the development of energy devices and perioperative management, the incidence of POPF remains between 17% and 40% [2,3,4]. Additionally, morbidity rates of POPFs reach up to 30% because of its potential to lead to intraabdominal bleeding or abscess [5], with the mortality rates of DP reaching approximately 5%, even in high-volume centers [6]. Recent evidence showed that variables such as obesity, estimated blood loss, nutritional status, and surgical methods for pancreatic resection are clinical predictors of POPF after DP [7,8,9]. Additionally, more recent reports showed that several surgical methods, including spraying fibrin glue, wrapping hydrogel [10] or a polyglycolic acid sheet [11], and using fibrin sealant [12], could reduce the incidence of POPF. In contrast, recent studies concluded that POPF occurrence could not be predicted using any clinical variables [13] and found that reinforced staplers did not reduce POPF incidence [14]. Thus, there is an urgent need to identify more robust factors that may help predict the risk of POPF. The *C*-reactive protein (CRP)-to-albumin ratio (CAR) was initially developed as a prognostic factor for patients with sepsis [15]. However, many studies showed that CAR is associated with prognosis for patients in several types of cancers, including pancreatic cancer [16,17,18]. Moreover, a recent meta-analysis revealed that CAR becomes a predictive factor for pancreatic cancer patients [19]. On the other hand, CAR can affect POCs such as anastomotic leakage in esophageal and colorectal surgery [20,21]. Considering this evidence, we hypothesized that CAR can predict not only prognosis but also POCs such as POPF. In addition, based on the relationship between POPF and malnutrition, this study aimed to determine whether CAR could be a potential predictor of POPF in patients who underwent DP for pancreatic cancer.

## 2. Materials and Methods

### 2.1. Patients

Between January 2009 and August 2022, 72 patients underwent DP for pancreatic cancer at Ehime University Hospital. We retrospectively analyzed the medical records of these patients. The inclusion criteria were as follows: (1) pancreatic cancer patients with preoperative or postoperative pathological diagnosis, (2) cases with resectable pancreatic cancer, and (3) patients with a tolerance for curative surgery. The exclusion criteria were as follows: (1) non-radical resection, (2) DP with celiac artery resection, and (3) peritoneal dissemination. However, the presence of neoadjuvant therapy such as chemotherapy and radiation was not included in the exclusion criteria. The study protocol was reviewed and approved by the ethics committee of the Ehime University Hospital in 2022. All patients or their guardians had verbally consented to use their medical information for scientific research (Ethics approval number: 2206005). Obtaining informed consent from all patients was waived because of the retrospective nature of the study. All patients underwent DP with splenectomy and lymph node dissection, with the closure of the pancreatic remnant performed using a stapler. The drainage tube was placed into the subphrenic space or pancreatic stump, depending on the surgeon’s decision.

### 2.2. Clinicopathological Data

The following data were collected from medical records: occurrence of POPFs, demographic variables (sex and age), anthropometric parameters (height, weight, and BMI), comorbidities, American Society of Anesthesiologists (ASA)’s physical status classification, blood transfusions, estimated blood loss, operative time, and serum albumin levels. POPFs were classified according to the International Study Group of Pancreatic Fistula (ISGPF) definition and grading [22]. In this study, grade B and higher indicated clinically relevant POPFs, which are symptomatic and require interventions such as antibiotics therapies or drainage for grade B and resuscitation or exploratory laparotomy for grade C fistulas. Drain amylase was monitored on postoperative days 1, 3, 5, and 7.

### 2.3. Nutritional Assessment Using CAR

CAR was calculated as *CAR =* [*CRP* (mg/dL)]/[*albumin* (g/dL)]. This calculation method was applied regardless of sex in the same way [15]. 

### 2.4. Statistical Analysis

All statistical analyses were performed using SPSS, version 24 (SPSS Inc., Chicago, IL, USA). Differences between patients with and without POPFs were compared using Mann–Whitney’s U test, Fisher’s exact test, or a chi-squared test. Additionally, patients’ backgrounds were expressed as the median and interquartile ranges for nonparametric distribution. The chosen cutoff value of CAR was based on a receiver operating characteristic (ROC) curve analysis using Youden’s index. Similarly, the cutoff values for continuous variables were calculated using their respective ROC curves. The potential risk factors for POPFs were evaluated using univariate and multivariate analyses. Univariate analysis was conducted using the chi-squared or Fisher’s exact test, followed by multivariate analysis using logistic regression to identify risk factors for POPFs. The results are presented as odds ratios and 95% confidence intervals. *p* values < 0.05 were considered to indicate statistical significance.

## 3. Results

### 3.1. Patient Characteristics

Among the 72 patients included, 35 were men and 37 were women. The median age was 71 (range 42–87) years. POPFs occurred in 17 (23.6%) patients. There was no mortality due to POPFs in this study. There were no statistically significant differences between patients with POPFs and those without POPFs with respect to age, sex, ASA classification, neoadjuvant chemotherapy, surgical approach method, and diabetes mellitus. However, preoperative albumin, CRP, and CAR were significantly higher in patients with POPFs than in those without (*p* = 0.001) (Table 1). Additionally, estimated blood loss, blood transfusions, the presence of a soft pancreas, and CD classification over III showed no significant difference. In contrast, the operation time was statistically significant (Table 2).

### 3.2. Calculation of Optimal CAR Cutoff Value 

The ROC analysis showed that the areas under the curve of albumin, CRP, and CAR were 0.669, 0.866, and 0.888, respectively (Figure 1). Thus, CAR was a better predictive marker for POPFs following DP. Using the Youden index, a CAR of 0.05 was determined to be the appropriate cutoff value, with a sensitivity of 76.5%, a specificity of 88.9%, and a likelihood ratio of 6.88. Patients were categorized into two groups based on the CAR cutoff value: the High-CAR group (CAR ≥ 0.05, *n* = 21) and the Low-CAR group (CAR < 0.05, *n* = 51). POPFs were observed in 61.9% of patients in the High-CAR group and 7.8% in the Low-CAR group. Univariate analysis was performed to evaluate whether a CAR ≥ 0.05 was a risk factor for POPFs after DP (*p* < 0.001). Similarly, multivariable logistic regression analysis revealed that a CAR ≥ 0.05 was an independent predictor of POPFs following DP (*p* = 0.013) (Table 3).

## 4. Discussion

POCs following pancreatectomy, including POPF, may worsen patient prognosis [22,23,24,25,26]. The incidence of POPF was approximately 21–40% in patients who underwent DP [4,27]. Several surgical techniques for pancreatic stump creation or pancreatic transection have been introduced to reduce the risk for POPF [4,5,28]. However, no robust evidence has been established to support surgical techniques. In contrast, a number of POPF risk factors have been suggested, such as a soft pancreas, obesity, diabetes mellitus, a lower geriatric nutritional risk index (GNRI), lower albumin levels, blood loss, and an extended operation time [29,30,31]. Notably, a meta-analysis revealed that a soft pancreas, a higher BMI, blood transfusion, blood loss, and the operative time were major predictors of POPF [7]. Especially, BMI is a well-known risk factor for POPF following pancreatectomy, as the alternative fistula risk score for pancreatoduodenectomy includes BMI as one of the assessments [32,33]. In the present study, those parameters actually showed a statistical relationship in the univariate analysis. However, recent data contrastingly indicated that definitive indicators for predicting POPF did not exist [13]. Therefore, exploring more reliable factors for POPF is an important point of clarification for surgeons. Lower albumin, including malnutritional status, is the most commonly reported risk factor for POPF after pancreatectomy [7,34,35]. Giardino et al. showed that preoperative elevated CRP levels were associated with an increased risk of POCs after pancreatectomy [36]. Under these circumstances, we hypothesized that preoperative CAR could be a novel predictor for POPF following DP. It is important to perform surgery based on preoperative POPF risk because POPF may result in increased medical costs and worsened patient prognosis [25,37]. Given these clinical issues, a parameter or strategy for simple preoperative assessment is needed. Recent reports revealed that some parameters using nutritional status and inflammation might contribute to the development of POCs following pancreatectomy, including the prognostic nutritional index [37], neutrophil-to-lymphocyte ratio [38,39], GNRI, and CAR [25]. Moreover, Gililland et al. suggested that albumin levels < 2.5 mg/dL or weight loss >10% warranted the postponement of surgery to improve operative outcomes [40]. Preoperative immunonutrition has also been reported to improve the outcomes of patients with pancreatic cancer [41,42] and reduce the risk for POPF [43].

CAR was originally developed to predict prognosis in patients with sepsis [15]. In this study, 23.6% of patients developed POPF following DP. CAR ≥ 0.05 was associated with an increased POPF risk, suggesting that the preoperative improvement of nutrition or inflammatory status might decrease POPF incidence. Our results also showed that nutritional or inflammatory status affected the risk of POCs, which was consistent with the findings of previous studies [25,44,45,46,47]. Previous data revealed that the CAR on postoperative day 3 is a risk factor for POPFs following pancreatoduodenectomy [48,49]. By the ISGPF definition of POPF, a POPF is diagnosed on postoperative day 3 due to the drain amylase level. For surgeons, the risk of POPF should be known preoperatively to perform safe procedures.

This study had a few limitations. First, the sample size was small, and only a single-center study was conducted to definitively claim that preoperative CAR was a novel POPF risk factor. Second, the retrospective nature of this study was another limitation of the scope of the conclusions. Finally, the level of CRP has the potential to depend on several factors including sex, body weight, and race [50]. Therefore, a larger prospective study should be conducted to validate this result. Despite these limitations, we believe that the predictor will be simple and valuable as a clinical application.

## 5. Conclusions

This study showed that a preoperative CAR ≥ 0.05 may become a risk factor for POPF following DP.

## Figures and Tables

**Figure 1 nutrients-14-05277-f001:**
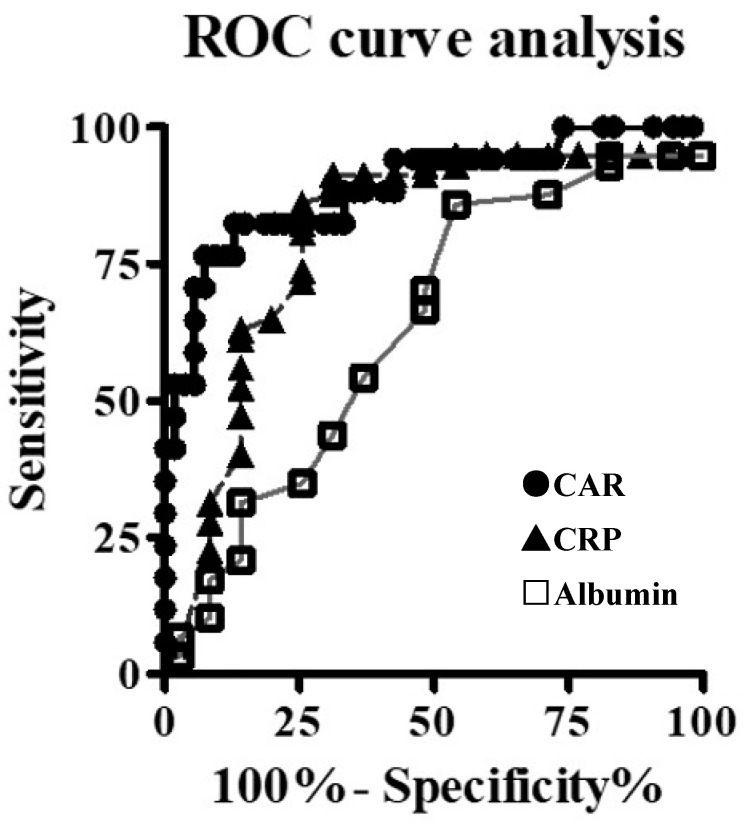
Comparison and determination of the *c*-reactive protein-to-albumin ratio cutoff value using the receiver operating characteristic curve analysis.

**Table 1 nutrients-14-05277-t001:** Preoperative variables in patients with and without POPFs following distal pancreatectomy (DP).

Preoperative Variables	POPF Group	Non-POPF Group	*p*-Value
	(N = 17)	(N = 55)	
Age (years)	66.6 (56–81)	70.9 (42–87)	0.129
Sex (male/female)	4/13	22/33	0.383
BMI	24.2 (19.2–28.9)	22.4 (16.3–29.0)	0.032
ASA classification			0.897
1 or 2	16 (94.1%)	48 (87.3%)	
3	1 (5.9%)	7 (12.7%)	
Neoadjuvant chemotherapy (%)	3 (17.6%)	9 (16.4%)	0.717
Surgical approach			
Laparotomy (%)	16 (94.1%)	49 (89.1%)	0.670
Laparoscopy (%)	1 (5.9%)	6 (10.9%)	
Preoperative			
HbA1c (%)	6.58 ± 0.21	6.65 ± 0.18	0.856
Total lymphocyte counts (×10^3^/μL)	1.48 ± 0.1	1.46 ± 0.1	0.930
Plt (×10⁴/μL)	18.89 ± 1.72	20.45 ± 0.91	0.411
CRP (mg/dL)	1.29 ± 0.64	0.10 ± 0.01	0.001
Albumin (g/dL)	3.62 ± 0.13	3.92 ± 0.06	0.027
CAR	0.35 ± 0.17	0.03 ± 0.01	0.001

POPF: postoperative pancreatic fistula; DP: distal pancreatectomy; BMI: body mass index; ASA: American Society of Anesthesiologists; CRP: *C*-reactive protein; CAR: CRP-to-Albumin ratio.

**Table 2 nutrients-14-05277-t002:** Intra- and postoperative variables in patients with and without POPFs.

Intra- and Postoperative	POPF Group	Non-POPF Group	*p*-Value
Variables	(N = 17)	(N = 55)	
Operation time (min)	473 (289–856)	344 (164–852)	0.001
Estimated blood loss (mL)	802 (35–3010)	451 (10–3360)	0.051
Blood transfusion (%)	3 (17.6)	8 (14.5)	0.778
Soft pancreas (%)	12 (70.6)	39 (70.9)	0.896
POCs excluding POPFs			
CD-grade over III (%)	2 (11.8)	8 (14.5)	0.753

POCs: postoperative complications; POPF: postoperative pancreatic fistula; POPF-related POCs: intraabdominal bleeding, surgical site infection; CD: Clavien-Dindo.

**Table 3 nutrients-14-05277-t003:** Univariate and multivariate analyses using logistical regression.

Parameters	Odds Ratio (95% CI)	*p*-Value	Odds Ratio (95% CI)	*p*-Value
BMI ≥ 23.8	4.354 (1.373–13.800)	0.020	1.605 (0.351–7.333)	0.542
Estimated blood loss ≥ 429 (g)	0.502 (0.258–0.977)	0.020	0.026 (0.001–0.033)	0.981
Operation time ≥ 374 (min)	5.224 (1.371–8.810)	0.009	2.190 (0.468–10.257)	0.320
CAR ≥ 0.05	0.046 (0.012–0.0326)	<0.001	12.419 (2.687–57.393)	0.013

CI: confidence interval; BMI: body mass index; CAR: *C*-reactive protein-to-albumin ratio.

## Data Availability

The data will be available upon request from the corresponding author.

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
