# Peer review of "Association between the Preoperative C-Reactive Protein-to-Albumin Ratio and the Risk for Postoperative Pancreatic Fistula following Distal Pancreatectomy for Pancreatic Cancer"

_nutrients, 2022, doi:10.3390/nu14245277_

Round 1
Reviewer 1 Report
While the presented research results are interesting and may have a significant application aspect, the downside is the small population of people taking part in the test.
The level of CRP depends on many factors, including gender, weight, race, age, and drugs used. Table 1 shows the influence of BMI. With a larger population, finding other correlations related to factors directly affecting CRP would be possible.
The conclusions should be extended.
Author Response
Reviewer's comment:
While the presented research results are interesting and may have a significant application aspect, the downside is the small population of people taking part in the test.
The level of CRP depends on many factors, including gender, weight, race, age, and drugs used. Table 1 shows the influence of BMI. With a larger population, finding other correlations related to factors directly affecting CRP would be possible. The conclusions should be extended.
We appreciate revierew's precious comment. We did not know that CRP depends on race, gender, and so on. So, we added this important opinion in disucussion as a limitation.
Reviewer 2 Report
The objective of this article was to investigate whether CAR could be used as an indicator for the risk of pancreatic fistula after distal pancreatectomy for pancreatic cancer. This study analyzed 72 pancreatic cancer patients treated at Ehime University. However, I am concerned about the following questions.
1. The most important problem is the selection of the number of patients. The sample size is too small to support the conclusion. The current data only indicate that CAR may be associated with postoperative pancreatic fistula. In order to obtain more generalization value and convincing results, the number and diversity of samples should be increased.
2. The second problem is the selection of samples. The authors exclude the difference of gender, because the proportion of gender distribution is almost the same. However, the age difference cannot be ruled out. After all, the sample size is small, so it is impossible to draw a completely accurate conclusion, nor can it be said that the age difference can be ruled out.
3. Another thing to note is the difference in BMI between the POPF group and Non-POPF group, which is an interesting finding. It is best to reflect this finding in the article.
4. The author can modify the introduction. In the description of POC in line 47, the twist is too blunt to subtly lead to the conjecture that CAR might predict POPF. It is best to cite the literature, expand the description, and combine the results of this study, the conclusion and the imagination of future research.
Author Response
Reviewer's comment:
The objective of this article was to investigate whether CAR could be used as an indicator for the risk of pancreatic fistula after distal pancreatectomy for pancreatic cancer. This study analyzed 72 pancreatic cancer patients treated at Ehime University. However, I am concerned about the following questions.
- The most important problem is the selection of the number of patients. The sample size is too small to support the conclusion. The current data only indicate that CAR may be associated with postoperative pancreatic fistula. In order to obtain more generalization value and convincing results, the number and diversity of samples should be increased.           ⇒We are really sorry for that. In order to get more numbers, we are planning to perform multicenter study as a next step.
- The second problem is the selection of samples. The authors exclude the difference of gender, because the proportion of gender distribution is almost the same. However, the age difference cannot be ruled out. After all, the sample size is small, so it is impossible to draw a completely accurate conclusion, nor can it be said that the age difference can be ruled out.                               ⇒We included the difference of gender in Table 1. But, we had descrived the sex as % female. So sorry if description content left reviewer confused. We changed it as it is easy for reader to understand what we meant.
- Another thing to note is the difference in BMI between the POPF group and Non-POPF group, which is an interesting finding. It is best to reflect this finding in the article.                      ⇒We are grateful for reviewer's comment. As reviewer pointed out, BMI is well known risk factor for POPF. We added sentence in discussion and also added refferences.  
- The author can modify the introduction. In the description of POC in line 47, the twist is too blunt to subtly lead to the conjecture that CAR might predict POPF. It is best to cite the literature, expand the description, and combine the results of this study, the conclusion and the imagination of future research.                          ⇒According to revewer's recommendation, we re-wrote introduction and added 6 reffernces to enhance our research significance. We really appreciate this comment to improve the quality of our manuscript.
Round 2
Reviewer 2 Report
The revised manuscript is more rigorous, but still needs to solve the biggest problem of the article, which is the small number of samples. The author proposes to carry out further research in the future and looks forward to the completion of this work as soon as possible.